# Urban–Rural Disparities in Case Fatality of Community-Acquired Sepsis in Germany: A Retrospective Cohort Study

**DOI:** 10.3390/ijerph20105867

**Published:** 2023-05-18

**Authors:** Claudia T. Matthaeus-Kraemer, Norman Rose, Melissa Spoden, Mathias W. Pletz, Konrad Reinhart, Carolin Fleischmann-Struzek

**Affiliations:** 1Institute of Infectious Diseases and Infection Control, Jena University Hospital, 07743 Jena, Germany; claudia.matthaeus@uni-jena.de (C.T.M.-K.);; 2Center for Sepsis Control and Care, Jena University Hospital, 07747 Jena, Germany; 3Wissenschaftliches Institut der Ortskrankenkassen, 10178 Berlin, Germany; 4Department of Anesthesiology and Operative Intensive Care, Charité University Medicine Berlin, 10117 Berlin, Germany

**Keywords:** sepsis, rural, urban, health-care disparities, pollution, environment

## Abstract

Background: We aimed to examine urban–rural disparities in sepsis case fatality rates among patients with community-acquired sepsis in Germany. Methods: Retrospective cohort study using de-identified data of the nationwide statutory health insurance AOK, covering approx. 30% of the German population. We compared in-hospital- and 12-month case fatality between rural and urban sepsis patients. We calculated odds ratios (OR) with 95% confidence intervals and the estimated adjusted odds ratio (OR_adj_) using logistic regression models to account for potential differences in the distribution of age, comorbidities, and sepsis characteristics between rural and urban citizens. Results: We identified 118,893 hospitalized patients with community-acquired sepsis in 2013–2014 with direct hospital admittance. Sepsis patients from rural areas had lower in-hospital case fatality rates compared to their urban counterparts (23.7% vs. 25.5%, *p* < 0.001, Odds Ratio (OR) = 0.91 (95% CI 0.88, 0.94), OR_adj_ = 0.89 (95% CI 0.86, 0.92)). Similar differences were observable for 12-month case fatalities (45.8% rural vs. 47.0% urban 12-month case fatality, *p* < 0.001, OR = 0.95 (95% CI 0.93, 0.98), OR_adj_ = 0.92 (95% CI 0.89, 0.94)). Survival benefits were also observable in rural patients with severe community-acquired sepsis or patients admitted as emergencies. Rural patients of <40 years had half the odds of dying in hospital compared to urban patients in this age bracket (OR_adj_ = 0.49 (95% CI 0.23, 0.75), *p* = 0.002). Conclusion: Rural residence is associated with short- and long-term survival benefits in patients with community-acquired sepsis. Further research on patient, community, and health-care system factors is needed to understand the causative mechanisms of these disparities.

## 1. Introduction

Sepsis is a life-threatening organ dysfunction that arises from a dysregulated host response to infection [1]. In 2017, there were an estimated 11 million sepsis-related deaths worldwide, which is equivalent to 20% of global deaths [2]. In Germany, sepsis incidence was 314/100,000 in 2013, and 30.5% of pwatients died [3]. The risk of death from sepsis increases with every delay in source control and adequate antimicrobial treatment [4]. Thus, at least one out of ten sepsis deaths is considered preventable through early diagnosis and adequate management [5].

Several studies have found that patients living in rural areas have less or delayed access to health-care services such as primary and secondary ambulatory care, hospital care, and post-acute care [6,7,8]. For many medical emergencies such as stroke [9] and acute myocardial infarction [10], survival benefits were described for patients residing in urban areas compared to patients from rural areas. Furthermore, rural hospitals often treat lower case volumes, which were found to be associated with higher mortality rates in several acute diseases, including sepsis [11]. In line with these observations, an increased population-level sepsis incidence and mortality were observed in medically underserved areas in the US, which are defined by a composite score of a population’s age, economic status, and access to basic medical services and often comprise rural areas [11,12]. The authors hypothesized that limited access to health care in underserved areas might contribute to delays in sepsis management and pose barriers to the management of chronic comorbidities, thus leading to *higher* mortality rates [12]. However, this study found no difference in sepsis case fatality rates between patients residing and those not residing in medically underserved areas. Moreover, in the US, Detelich et al. examined a national sample of community-acquired sepsis patients and found no association between longer distances from home-to-hospital and poorer hospital outcomes [13]. In contrast, sepsis was the only medical emergency with a lower hospital case fatality rate in rural compared to urban patients in a large study on the urban–rural health gap based on the National Inpatient Sample in the US [14].

These results seem conflicting, and the underlying mechanisms are insufficiently understood. Furthermore, it is unclear if they can be transferred to the German health-care system. Therefore, we aimed to investigate the association between urban–rural residences and hospitals and the 12-month sepsis case fatality rate among patients with community-acquired sepsis in Germany, considering differences in patient and sepsis characteristics between urban and rural patients. These associations were assessed among all sepsis patients and subgroups at particular risk for urban–rural health gaps, such as elderly patients with potentially limited mobility and sepsis patients admitted as an emergency that may have required emergency medical service transport and emergency department treatment.

## 2. Materials and Methods

### 2.1. Study Design and Data Source

We used the data of the SEPFROK study, which investigated risk factors for developing severe sepsis or septic shock as well as burden of disease and sequelae of the sepsis in sepsis survivors. SEPFROK was a retrospective longitudinal cohort study based on de-identified data of the nationwide statutory health insurance AOK, which covers around 30% of the German population. The study was approved by the institutional review board of the Friedrich-Schiller-University Jena (2019-1282-Daten). The requirement for informed consent was waived because all data were de-identified. This study was reported according to the Strengthening the Reporting of Observational Studies in Epidemiology (STROBE) reporting guideline. The data source is described elsewhere in detail [15].

### 2.2. Patient Sample and Characteristics

The SEPFROK cohort consists of adult patients (>15 years) with hospital-treated sepsis between 1 January 2013 and 31 December 2014. Hospital-treated Sepsis identified explicit sepsis by ICD-10-German Modification (GM) primary and secondary discharge diagnosis among AOK beneficiaries (Appendix A). In 2013/2014, sepsis was defined according to the sepsis-1/2 criteria as systemic inflammatory response syndrome due to infection [16] in Germany and comprised patients with and without septic organ dysfunction (patients with and without severe sepsis according to the sepsis-1/2 definition). Explicit sepsis codes comprised clinical sepsis codes (“R codes”) and microbiological sepsis codes (e.g., “A codes” and “B codes”). Severe sepsis was defined by codes R56.1 (severe sepsis) and R57.2 (septic shock). Patients without continuous insurance status or sepsis in the 24 months prior to admission to the index hospitalization were excluded. For this study, we used a subsample of patients with community-acquired sepsis and primary hospital admission (excluding transfers from other hospitals, Figure 1).

Community-acquired sepsis was defined as absence of any ICD-10-GM code for nosocomial infections (Appendix A). A list of codes was derived from international literature, translated into ICD-10-GM, and reviewed by experts from controlling and infectious diseases. Direct admission included: non-emergency-admission (admission by physician) or admission as emergency. Patients transferred from other hospitals were excluded. Patients were classified as urban or rural patients according to the district types of settlement structure defined by the Federal Institute for Research on Building, Urban Affairs, and Spatial Development as included in the AOK data for each patient. We summarized urban and suburban regions as urban regions. The definition for each region is provided in Appendix A. We characterized patients regarding age, gender, comorbidities according to the Charlson Comorbidity Index [17], and pre-existing nursing care level (qualifying for monetary compensation for formal or informal nursing care in the German health-care system). Furthermore, we analyzed characteristics of sepsis (focus of infection, type of organ dysfunction), type of admission, and clinical treatments (surgical treatments, mechanical ventilation, dialysis). Definitions are provided in Appendix A.

### 2.3. Outcomes

We compared in-hospital case fatality and 12-month case fatality between rural and urban sepsis patients. Outcomes were analyzed for all patients and the following subgroups: age < 40 years, age 40–64 years, 65–80 years, age > 80, patients with severe sepsis and patients with hospital admission as emergency.

## 3. Statistical Analysis

The hospital and 12-month case fatality of sepsis in the rural and the urban population were estimated as the proportion of hospital and 12-month deaths among hospital sepsis cases in both populations. We report the crude risk difference (RD_crude_) and the odds ratio (OR_crude_), including standard errors and 95% confidence intervals, to describe the observed differences in the hospital and 12-month case fatality of sepsis between both populations.

Rural and urban cases cannot simply be assumed equal regarding patient and sepsis characteristics. Therefore, we also report adjusted risk differences (RD_adj_), the adjusted odds ratio (OR_adj_), and the adjusted relative risk (RR_adj_), taking age, gender, comorbidities, pre-existing nursing care dependency, pre-existing nursing home placement, the focus of infection, admission category (emergency vs. no emergency), and sepsis as the primary hospital discharge diagnosis (Appendix A) into account. It is important to note that the temporal order needs to be considered in the selection of covariates for adjusted group comparisons. Only covariates that are temporally prior or equal to the grouping variable of urban or rural residence can be confounding variables. Characteristics of sepsis, including the severity of sepsis, organ dysfunction, and measures of organ support, do not fulfill this temporal criterion. A person’s living environment exists before the onset of sepsis. The severity and the course of the sepsis can be a consequence of living in an urban or rural area but cannot be a confounder. Adjusting for posttreatment variables leads to biased parameter estimates [18]. Therefore, organ dysfunction and measures of organ support, for example, were not included in the adjustment model. For the calculation of the adjusted measures, we utilized a logistic regression model with the dichotomous indicator variable X_rural_ (with X_rural_ = 1 for sepsis cases from rural areas and X_rural_ = 0 for sepsis cases from urban areas) and all covariates. The estimated OR_adj_ is the exponential function of the logistic regression coefficient of the indicator variable X_rural_. RD_adj_ can be estimated from the logistic regression model by the average marginal effect (AME) [19]. The AME is equal to the difference π_rural_–π_urban_ of the average margins π_rural_ and π_urban_, which are the point estimates of the adjusted hospital case fatality rates of sepsis in both populations. We report the AME as well as the two average margins π_rural_ and π_urban_, with asymptotic standard errors, which were computed using the Delta method [19,20]. An additional Poisson regression model with robust standard errors was used for estimating RR_adj_ [21]. The model-based estimator of RR_adj_ is given by the exponential function of the regression coefficient of the indicator variable X_rural_. The asymptotic standard error of RR_adj_ was computed using the Delta method. Average margins, as well as marginal effects, were estimated using SAS/STAT^®^ and SAS/IML^®^ and the Margins macro (https://support.sas.com/kb/63/038.html; accessed on 10 May 2021). All other statistical computations were conducted with R, including the R packages DescTools (https://cran.r-project.org/web/packages/DescTools/index.html; accessed on 10 May 2021) and geepack (https://cran.r-project.org/web/packages/geepack/index.html; accessed on 10 May 2021).

## 4. Results

### 4.1. Patient Characteristics

Among 23.0 million AOK beneficiaries, we identified 118,893 hospitalized patients with community-acquired sepsis in 2013–2014 who were admitted directly to the hospital. Patient characteristics are shown in Table 1.

Rural patients were significantly older (75.0 vs. 74.1 years, *p* < 0.001) and more often female (49.7% vs. 49.0%, *p* = 0.032) compared to patients from urban areas. They had higher mean unweighted Charlson Comorbidity Index (CCI) scores and higher pre-sepsis nursing care levels (Table 1). Renal, cardiovascular diseases, diabetes, and dementia were more common among rural sepsis patients compared to urban sepsis patients, while pulmonary diseases, cancer, and HIV/AIDS affected fewer rural sepsis patients than urban sepsis patients. Rural sepsis patients were less frequently admitted as an emergency than their urban counterparts (53.8% vs. 66.1%; *p* < 0.001) and had slightly shorter mean hospital lengths of stay (15.97 vs. 16.24, *p* = 0.003). The occurrence of most organ dysfunctions was more frequent among urban sepsis patients; in particular, they had more often circulatory failure/shock.

### 4.2. Rural vs. Urban Outcomes

We found that sepsis patients from rural areas had lower in-hospital and 12 months case fatality rates compared to sepsis patients from urban areas (Table 2).

Adjusting for patient and sepsis characteristics, rural sepsis patients had similarly lower odds ratios for in-hospital and 12-month case fatality (OR_adj_ = 0.89 (95% CI 0.86, 0.92), *p* < 0.000, and OR_adj_ = 0.92 (95% CI 0.89, 0.94), *p* < 0.000, respectively, Table 3).

Lower crude and adjusted odds ratios for in-hospital and 12-month case fatality in favor of rural sepsis cases were also found in all age groups except for patients aged 40–64 years, for whom we did not observe any significant differences in case fatalities (Figure 2).

The lowest crude and adjusted OR for in-hospital case fatality were observed among rural patients < 40 years (OR = 0.51 (95% CI 0.32, 0.83), *p* = 0.008, OR_adj_ = 0.49 (95% CI 0.23, 0.75), *p* = 0.002). Crude and adjusted differences in 12-month case fatality were not statistically significant between rural and urban patients in the age groups < 40 years and 40–64 years. In the other age groups (65–80 years, >80 years), we also found survival benefits for rural sepsis patients. However, differences in 12-month case fatality between rural and urban patients were less pronounced compared to differences in hospital case fatality.

Rural patients with severe sepsis and sepsis patients with admittance as an emergency had similarly lower crude and adjusted odds ratios for in-hospital case fatality compared to patients with urban residence (severe sepsis: OR_adj_ = 0.87 (95% CI 0.83, 0.91), *p* < 0.001, emergency admission: OR_adj_ = 0.89 (95% CI 0.86, 0.93), *p* < 0.001). Differences in 12-month case fatality between rural vs. urban severe sepsis patients and sepsis patients admitted as an emergency were also evident but less pronounced (severe sepsis: OR_adj_ = 0.88 (95% CI 0.84, 0.93), *p* < 0.001, emergency admission: OR_adj_ = 0.93 (95% CI 0.90, 0.97), *p* < 0.001).

## 5. Discussion

Among 118,893 patients with community-acquired sepsis, we found that in-hospital hospital case fatality was lower among rural vs. urban patients (rural 23.7% vs. urban 25.5%, *p* < 0.001; *OR* = 0.91 (95% CI 0.88, 0.94)), also if adjusting for differences in patient- and infection-characteristics between groups (OR = 0.89 (95% CI 0.86, 0.92)). Such differences were also found regarding 12-month mortality but were less pronounced. Particularly patients < 40 years with community-acquired sepsis living in rural areas had half the odds of dying in a hospital compared to their urban counterparts (OR_adj_ = 0.49 (95% CI 0.23, 0.75), *p* = 0.002). Survival benefits were also found for rural patients with severe community-acquired sepsis or patients admitted as emergencies.

Our findings contradict the hypothesis that sepsis case fatality may be higher among rural patients, e.g., due to delays or barriers in access to care [12] and treatment in low-volume hospitals with potentially less experience in the treatment of sepsis [11]. They are in line with a previous study based on the US National Inpatient Sample, which found that sepsis was the only medical emergency with a lower hospital case fatality rate in rural compared to urban patients [14]. In this study, an OR of around 0.95 was found for in-hospital mortality comparing rural and urban patients.

The survival benefit observed for rural patients differs from observations made for other acute care diseases, such as stroke, for which no urban–rural disparities in case fatalities were found [22], and myocardial infarction, for which an urban survival benefit was described in Germany [23]. This raises the question of the underlying mechanism of this rural survival advantage in sepsis patients. We hypothesize that there may be protective factors on sepsis case fatality associated with rural residence. Urban citizens are increasingly bearing the burden of environmental risk factors such as a lack of green space [24], light pollution (especially artificial light at night) [25], and noise pollution [26], which could act as chronic stressors connected with immunomodulatory mechanisms. Recent studies confirmed in mouse as well as human models that chronic stress can comprise immunity, increase inflammatory reactivity [27], and is associated with higher risks of respiratory virus infections [28], reduced immunological responses to vaccines [29] as well as a higher risk of life-threatening infections (in case of stress-related disorders controlled for familial background and comorbidities) [30]. In line with this, Ojards et al. confirmed an association with stress, including chronic, low-grade inflammation and patients’ adjusted one- and ten-year incidence of sepsis [31]. Above that, there may be environmental factors with a direct influence on sepsis outcomes. In general, outdoor air pollution such as particulate matter (PM), gaseous pollutants (ozone, nitrogen dioxide, and sulfur dioxide), and mixed traffic-related air pollution were named as a major source of more than 3% of the annual disability-adjusted life years lost in the 2010 Global Burden of Disease comparative risk assessment [32]. Specifically, higher loads of tropospheric ozone are associated with increased lung inflammation, more respiratory symptoms, and higher morbidity as well as mortality rates [33]. Accordingly, previous studies reported a higher risk for sepsis mortality in dependence of increased levels of ozone [34]. Results regarding the impact of air temperature in combination with air pollution such as ozone, nitrogen dioxide (NO_2_) and PM_10_ on sepsis mortality were ambivalent [35].

Above these environmental factors, early sepsis treatment rests at the timely start of an adequate antibiotic and fluid therapy [36] and does mostly not require specified treatment units such as stroke units or cardiac catheter laboratories [22,23]. Therefore, a rural residence may not put patients at a disadvantage regarding their emergency care and may explain differences in urban–rural disparities found compared to stroke or myocardial infarction patients in Germany. Furthermore, differences in accessibility of care between urban and rural regions may be less pronounced in Germany. Previous research found only weak correlations between access to care of internal medicine, neurology, and surgery inpatient care and urbanity (*r*  =  0.31, *p*  <  0.001) [37]. Access to intensive care is also considered equally good all over Germany [38]; Germany has the highest accessibility index per 100,000 population, with an average travel time of 9.3 min to the next hospital. Smaller delays in hospital treatment may not significantly impact hospital mortality in patients with community-acquired sepsis [39]

Our study has several strengths. We studied a large, unselected cohort of sepsis patients identified in data of one of the largest German health-care providers covering 30% of the German population. We used a comprehensive adjustment considering differences in patient demographics, burden of comorbidities, and infection characteristics between rural and urban patients. Pre-existing comorbidities were identified in a 12-month look-back period prior to sepsis in inpatient and outpatient data.

The following limitations need to be considered. First, the identification of sepsis cases by explicit ICD codes suffers particularly from low sensitivity compared to clinical sepsis diagnosis in patients’ medical records [40]. Coding could be influenced by external factors in the DRG system. Generally, comparisons with clinical sepsis diagnoses in (electronic) health records showed that a considerable proportion of sepsis cases is missed in administrative data, which may also impact the estimated urban–rural disparities in sepsis case fatality. Likewise, risk factors could be coded with limited validity and thus impact the risk adjustment. Second, the sample of AOK patients may not be fully representative of the total German population. However, prior studies have suggested only small differences between AOK and non-AOK beneficiaries in Germany [41]. Third, we cannot rule out a residual bias in the adjusted estimates due to differences in unmeasured confounders between the urban and rural patient groups, which could not be taken into account in our statistical analyses. Fourth, our data do not include information on hospital providers; thus, we cannot determine if patients were treated in urban or rural hospitals and conclude any associations with case fatality. More studies are needed to further explore the impact of provider characteristics and treatment-seeking behavior of patients on sepsis mortality and how they may contribute to the rural survival benefit we observed in Germany. Fifth, the sepsis cohort was identified in the years 2013 and 2014. Possible medical and environmental changes may limit the applicability of the findings to current care settings and underscore the need to replicate the observations of this study in the context of current research.

## 6. Conclusions

Rural residency is associated with short-term and long-term survival benefits in patients with community-acquired sepsis.

Rural residency is associated with short-term and long-term survival benefits in patients with community-acquired sepsis. Given that our world is becoming increasingly urbanized, with an estimated 7 out of 10 people in the world that will live in urban areas by 2050 [42], and the United Nations (UN) identifying urbanization as one of four “demographic mega-trends” [43], further research on patient, community, and health-care system factors that distinguish rural from urban patient populations is needed to understand the underlying mechanisms. This may also help to identify preventive measures to reduce the case fatality of sepsis and its long-term impact on patients, relatives and the society.

## Figures and Tables

**Figure 1 ijerph-20-05867-f001:**
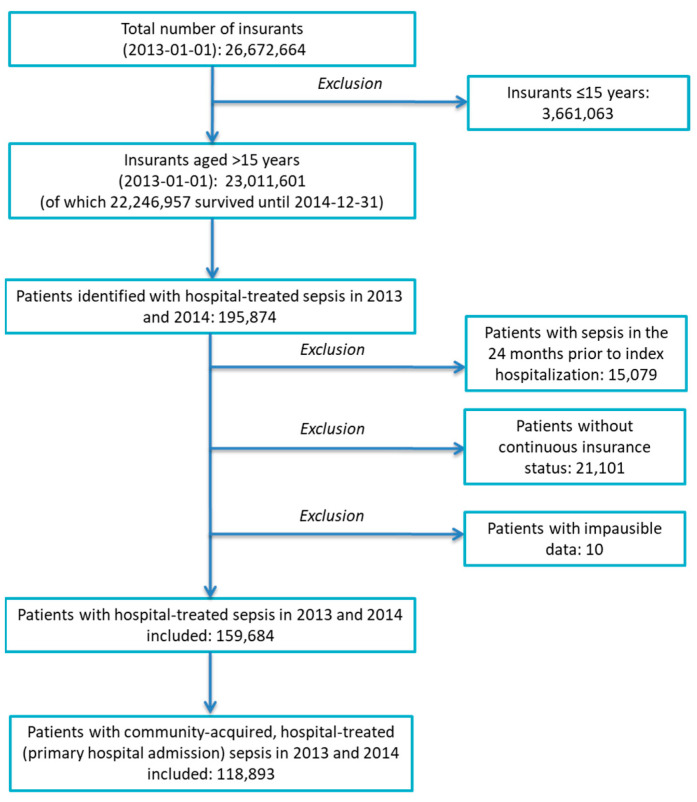
Flow of cohort.

**Figure 2 ijerph-20-05867-f002:**
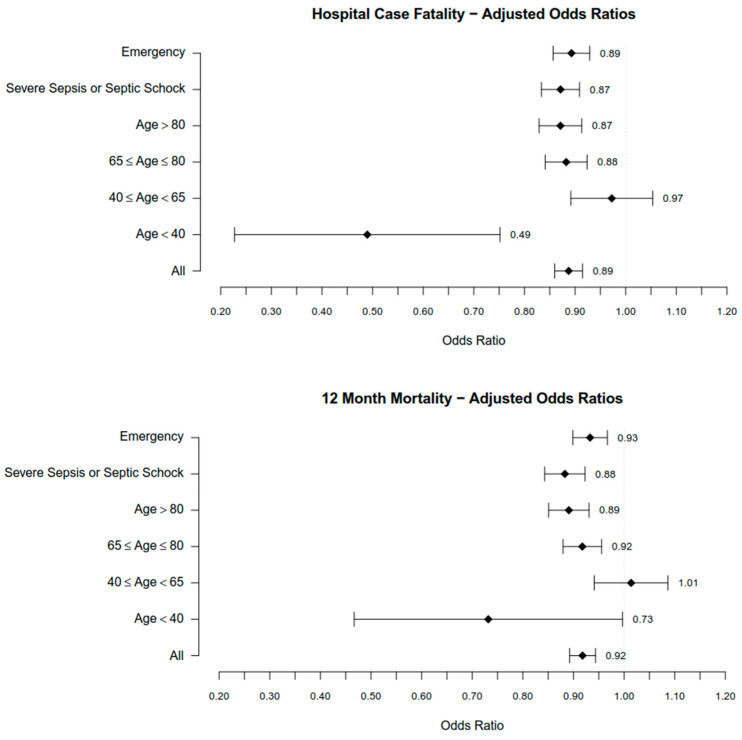
Rural–urban Hospital Case Fatality and 12-month case fatality (adjusted OR).

**Table 1 ijerph-20-05867-t001:** Patient characteristics and clinical features of patients with community-acquired sepsis by region.

	Urban	Rural	*p* Value (Unadjusted)	OR [95%-CI]
Number of observations	60,386 (50.8%)	58,507 (49.2%)		
Age in years ^m^	74.10 (13.07), 77 (15)	74.95 (12.44), 77 (15)	<0.001	
Female gender ^c^	41,121 (49.0%)	17,386 (49.7%)	0.032	1.03 [1.00, 1.05]
Nursing care level ^c^				
0	51,093 (60.9%)	20,583 (58.8%)	<0.001	0.92 [0.90, 0.94]
2	11,814 (14.1%)	5042 (14.4%)	0.138	1.03 [0.99, 1.07]
3	10,168 (12.1%)	4484 (12.8%)	0.001	1.07 [1.03, 1.11]
4	7459 (8.9%)	3293 (9.4%)	0.004	1.07 [1.02, 1.11]
5	3372 (4.0%)	1585 (4.5%)	<0.001	1.13 [1.07, 1.21]
**Charlson Comorbidity Index (CCI)**				
Charlson Comorbidity Index ^m^	3.27 (2.19), 3 (3)	3.48 (2.25), 3 (3)	<0.001	
Chronic pulmonary disease ^c^	27,884 (33.2%)	11,125 (31.8%)	<0.001	0.94 [0.91, 0.96]
Renal disease ^c^	27,096 (32.3%)	12,978 (37.1%)	<0.001	1.24 [1.20, 1.27]
HIV or AIDS ^c^	145 (0.2%)	25 (0.1%)	<0.001	0.41 [0.27, 0.63]
Diabetes ^c^	37,187 (44.3%)	16,808 (48.0%)	<0.001	1.16 [1.13, 1.19]
Congestive heart failure and myocardial infarction ^c^	33,942 (40.5%)	15,540 (44.4%)	<0.001	1.18 [1.15, 1.21]
Cancer ^c^	21,581 (25.7%)	8539 (24.4%)	<0.001	0.93 [0.91, 0.96]
Dementia or cerebrovascular disease ^c^	31,700 (37.8%)	14,362 (41.0%)	<0.001	1.15 [1.12, 1.18]
Liver disease ^c^	14,647 (17.5%)	6102 (17.4%)	0.955	1.00 [0.97, 1.03]
**Focus of infection** ^c^				
Respiratory ^c^	27,012 (32.2%)	11,229 (32.1%)	0.746	1.00 [0.97, 1.02]
Abdominal ^c^	9011 (10.8%)	3693 (10.6%)	0.355	0.98 [0.94, 1.02]
Wound / soft tissue ^c^	4790 (5.7%)	2256 (6.5%)	<0.001	1.14 [1.08, 1.20]
Urogenital ^c^	25,539 (30.5%)	11,000 (31.4%)	0.001	1.05 [1.02, 1.08]
Central nervous system ^c^	647 (0.8%)	258 (0.7%)	0.567	0.96 [0.83, 1.11]
Device-related ^c^	2491 (3.0%)	1049 (3.0%)	0.800	1.01 [0.94, 1.09]
Pregnancy-associated ^c^	61 (0.1%)	21 (0.1%)	0.524	0.83 [0.50, 1.36]
**Organ Dysfunctions**				
Circulatory/shock ^c^	11,351 (13.5%)	4055 (11.6%)	<0.001	0.84 [0.81, 0.87]
Respiratory ^c^	31,459 (37.5%)	13,173 (37.7%)	0.613	1.01 [0.98, 1.03]
Encephalopathy ^c^	11,308 (13.5%)	4590 (13.1%)	0.100	0.97 [0.93, 1.01]
Renal dysfunction ^c^	23,286 (27.8%)	9098 (26.0%)	<0.001	0.92 [0.89, 0.94]
Metabolic acidosis ^c^	6795 (8.1%)	2602 (7.4%)	<0.001	0.91 [0.87, 0.96]
Hepatic ^c^	2196 (2.6%)	822 (2.4%)	0.008	0.90 [0.83, 0.97]
Thrombocytopenia ^c^	6730 (8.0%)	2611 (7.5%)	0.001	0.93 [0.88, 0.97]
Number of organ dysfunctions	1.110 (1.225), 1 (2)	1.056 (1.186), 1 (2)	<0.001	
**Type of admission** ^c^				
Non-emergency admission ^c^	28,407 (33.9%)	16,167 (46.2%)	<0.001	1.68 [1.64, 1.72]
Admission as emergency ^c^	55,499 (66.1%)	18,820 (53.8%)	<0.001	0.60 [0.58, 0.61]
Surgical treatment ^c^	23,144 (27.6%)	9596 (27.4%)	0.588	0.99 [0.97, 1.02]
ICU treatment	23,243 (27.7%)	8658 (24.8%)	<0.001	0.86 [0.83, 0.88]
Mechanical ventilation ^c^	1684 (20.1%)	6433 (18.4%)	<0.001	0.90 [0.87, 0.93]
Renal replacement therapy ^c^	6222 (7.4%)	2477 (7.1%)	0.044	0.95 [0.91, 1.00]
Hospital length of stay in days ^m^	16.24 (14.88), 12 (12)	15.97 (14.57). 12 (12)	0.003	

Note: ^m^ Descriptive statistics of metric variables are *M* (*SD*), *Med* (*IQR*). The *p*-value refers to the Null hypothesis of no mean differences between rural and urban cases. ^c^ Descriptive statistics of categorical variables are *n* (*h*), with *n* the absolute frequencies and *h* the relative frequencies in%. The *p*-value refers to the Null hypothesis of equal probabilities between rural and urban cases.

**Table 2 ijerph-20-05867-t002:** Crude statistical comparisons of hospital cases fatality and 12-month case fatality in urban and rural sepsis cases by subgroups.

		Urban		Rural		Crude Group Comparisons			
Outcome	Group	*n*	Death *n* (%)	*n*	Death *n* (%)	*p*	OR [95% CI]	RR [95% CI]	Risk Difference [95% CI]
Hospital case fatality	all	83,906	21,386 (25.5%)	34,987	8293 (23.7%)	<0.001	0.91 [0.88, 0.94]	0.93 [0.91, 0.95]	−1.79 [−2.32, −1.25]
	<40	1689	123 (7.3%)	515	20 (3.9%)	0.008	0.51 [0.32, 0.83]	0.53 [0.34, 0.85]	−3.40 [−5.34, −1.09]
	40–64	15,315	2909 (19.0%)	5871	1061 (18.1%)	0.128	0.94 [0.87, 1.02]	0.95 [0.89, 1.01]	−0.92 [−2.07, 0.25]
	65–80	37,728	9316 (24.7%)	15,625	3551 (22.7%)	<0.001	0.90 [0.86, 0.94]	0.92 [0.89, 0.95]	−1.97 [−2.75, −1.17]
	>80	29,174	9038 (31.0%)	12,976	3661 (28.2%)	<0.001	0.89 [0.84, 0.92]	0.91 [0.88, 0.94]	−2.77 [3.70, −1.82]
	severe sepsis	33,647	15,597 (46.4%)	13,623	5943 (43.6%)	<0.001	0.90 [0.86, 0.93]	0.94 [0.92, 0.96]	−2.73 [−3.72, −1.74]
	admission as emergency	55,499	14,477 (26.1%)	18,820	4623 (24.6%)	<0.001	0.92 [0.89, 0.96]	0.94 [0.92, 0.97]	−1.52 [−2.23, −0.80]
12 months case fatality	all	83,906	39,442 (47.0%)	34,987	16,014 (45.8%)	<0.001	0.95 [0.93, 0.98]	0.97 [0.96, 0.99]	−1.24 [−1.86, −0.61]
	<40	1689	224 (13.3%)	515	55 (10.7%)	0.142	0.78 [0.57, 1.07]	0.81 [0.61, 1.16]	−2.58 [−5.53, 0.75]
	40–64	15,315	5376 (35.1%)	5871	2034 (34.6%)	0.542	0.98 [0.92, 1.04]	0.99 [0.95, 1.03]	−0.46 [−1.88, 0.98]
	65–80	37,728	17,166 (45.5%)	15,625	6826 (43.7%)	<0.001	0.93 [0.90, 0.97]	0.96 [0.94, 0.98]	−1.81 [−2.74, −0.89]
	>80	29,174	16,676 (57.2%)	12,976	7099 (54.7%)	<0.001	0.91 [0.87, 0.94]	0.96 [0.94, 0.98]	−2.45 [−3.48, −1.43]
	severe sepsis	33,647	21,381 (63.6%)	13,623	8402 (61.7%)	<0.001	0.92 [0.89, 0.96]	0.97 [0.96, 0.99]	−1.87 [−2.84, −0.91]
	admission as emergency	55,499	26,076 (47.0%)	18,820	8677 (46.1%)	0.037	0.97 [0.93, 1.00]	0.98 [0.96, 1.00]	−0.88 [−1.70, -0.06]

**Table 3 ijerph-20-05867-t003:** Covariate-adjusted statistical comparisons of hospital cases fatality and 12-month case fatality in urban and rural sepsis cases by subgroups.

		Marginal Predictive Means (Adjusted Proportion of Death%)		Adjusted Group Comparisons			
Outcome	Subgroup	Urban	Rural	*p*	OR_adj_ [95% CI]	RR_adj_ [95% CI]	AME ^1^ [95% CI]
Hospital case fatality	all	25.55 [25.27, 25.83]	23.56 [23.14, 23.99]	<0.001	0.89 [0.86–0.92]	0.92 [0.90–0.94]	−1.99 [−2.50,–1.48]
	<40	7.21 [6.07, 8.34]	4.04 [2.38, 5.70]	0.002	0.49 [0.23–0.75]	0.56 [0.30–0.82]	−3.17 [−5.21,–1.13]
	40–64	18.84 [18.26, 19.43]	18.46 [17.51, 19.42]	0.510	0.97 [0.99–1.05]	0.98 [0.92–1.04]	−0.38 [−1.51–0.75]
	65–80	24.72 [24.30, 25.13]	22.67 [22.04, 23.30]	<0.001	0.88 [0.84–0.92]	0.92 [0.89–0.95]	−2.05 [−2.81, −1.29]
	>80	30.93 [30.43, 31.44]	28.32 [27.57, 29.06]	<0.001	0.87 [0.83–0.91]	0.92 [0.89–0.94]	−2.62 [−3.53, −1.71]
	severe sepsis	46.44 [45.93, 46.94]	43.44 [42.65, 44.22]	<0.001	0.87 [0.83–0.91]	0.94 [0.92–0.96]	−3.00 [−3.93, −2.06]
	admission as emergency	26.19 [25.84, 26.54]	24.27 [23.69, 24.85]	<0.001	0.89 [0.86–0.93]	0.93 [0.90–0.95]	−1.92 [−2.60, −1.24]
12 months case fatality	all	47.16 [46.85, 47.47]	45.41 [44.93, 45.88]	<0.001	0.92 [0.89, 0.94	0.96 [0.95, 0.98]	−1.76 [−2.33, −1.18]
	<40	13.27 [11.83, 14.71]	10.69 [8.26, 13.11]	0.076	0.73 [0.47, 1.00]	0.82 [0.60, 1.04]	−2.58 [−5.43, 0.27]
	40–64	34.901 [34.23, 35.58]	35.16 [34.06, 36.26]	0.707	1.01 [0.94, 1.09]	1.01 [0.97, 1.05]	0.25 [−1.05, 1.55]
	65–80	45.49 [45.03, 45.95]	43.71 [43.00, 44.43]	<0.001	0.92 [0.88, 0.96]	0.96 [0.94, 0.98]	−1.78 [−2.63, −0.92]
	>80	57.14 [56.65, 57.72]	54.66 [53.84, 55.47]	<0.001	0.89 [0.85, 0.93]	0.96 [0.94, 0.97]	−2.53 [−3.51, −1.54]
	severe sepsis	63.72 [63.24, 64.19]	61.26 [60.50, 62.01]	<0.001	0.88 [0.84, 0.93]	0.96 [0.95, 0.98]	−2.46 [−3.36, −1.56]
	admission as emergency	47.13 [46.75, 47.50]	45.70 [45.05, 46.35]	<0.001	0.93 [0.90, 0.97]	0.97 [0.95, 0.99]	−1.43 [−2.18, −0.68]

Note: ^1^ AME = Average Marginal Effect.

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
