# Peer review of "Urban–Rural Disparities in Case Fatality of Community-Acquired Sepsis in Germany: A Retrospective Cohort Study"

_ijerph, 2023, doi:10.3390/ijerph20105867_

Round 1
Reviewer 1 Report
The manuscript describes the short- and long-term case fatality of rural vs urban septic patients in a large population study from Germany. The topic is timely and of substantial importance for clinical care and health policy efforts to reduce sepsis-related outcome disparities. The manuscript is well-written. However, there are several areas of concern.
Major comments
1. The reported cohort includes septic patients managed nearly 10 years ago. Progress in patient care and potential population/environmental changes may limit the generalizability of reported findings to contemporary rural vs urban sepsis outcomes. This consideration should be included under study limitations.
2. Lines 54-55: the cited study by Goodwin et al had indeed similar case fatality among septic patients living in medically underserved areas vs those not residing in such areas; however, both crude and adjusted mortality rates (which reflect better a disease-specific population burden) were markedly higher among the former.
3. Lines 108-110, 118-120, and 289-290: the authors collected data on type of organ dysfunction as well as on organ support (mechanical ventilation and hemodialysis). However, the latter 2 categories were not included in the adjusted models, as noted in lines 118-120, with this selection confirmed in the Discussion (lines 289-220). Illness severity is a key confounder in examination of patient outcomes and would be expected to affect the estimates of the association of rural vs urban residence with morality in sepsis. Because physiological data, commonly used in severity of illness scores based on clinical records, may not be available in administrative data, investigators often use organ dysfunction data and organ support interventions as proxies for severity of illness sin adjusted models. For example, a recent report by Bosch NA, et al (AnnalsATS 2022; 19(6): 1072-1076) demonstrated similar accuracy in predicting hospital mortality between the number of organ dysfunctions and SOFA scores. Model findings for in-hospital- and 12-month mortality overall and in the examined subgroups, adjusted for measure of illness severity should be reported. Summary data on the number of organ dysfunctions should be added to Table 1 for better perspective on the cohort.
4. How many patients in each group were admitted to ICU overall and among those with severe sepsis?
5. Lines 115-121: The risk-adjustment covariates used in modeling should be moved to the statistical analysis subsection by the relevant models for clarity.
6. Shorten the Results section and limit duplication of Tabular data in the narrative.
7. Lines 238-246: The implied comparable access to care between rural and urban populations in Germany does not explain the prognostic advantage of septic rural patients and would rather suggest an otherwise similar outcome.
8. Lines 253-254: the referenced study #25 is cited by the authors to indicate that in Germany there are “inverse associations between the number of hospital beds and sepsis outcomes” to support their postulate that septic patients treated in tertiary centers and university hospitals (presumably mostly urban patients) would have higher mortality and thus would explain in part the lower case fatality among rural patients. However, the quoted statement indicates the opposite, that is, that higher number of beds would be associated with lower mortality. More critical, however, is that the cited study by Engel et al showed no association between the number of beds or university hospitals and mortality on adjusted analyses. Thus, the prognostic advantage of rural septic patients in the present study Is not explained by the Discussion in lines 247-264. In addition, as noted by the authors in lines 262-264, some septic rural patients may have actually worse outcome when seeking treatment in tertiary centers, which would lead to the opposite effect than that observed in the present study, at least for some rural patients. The authors acknowledge the lack of data on hospital characteristics in their data under study limitations.
9. Lines 265-286: the authors postulate higher burdens of environmental risk factors among urban residents as an explanation for the lower mortality among septic rural residents, including among others, higher levels of chronic inflammation. If these are key factors, rural residents would be expected to have better outcomes for other acute conditions. This is, however, not the case, as the authors cite worse outcomes of rural patients for other acute conditions (lines 44-45) in other countries (mostly high-income ones), which appear at odds with the proposed argument. Notably, the rural residents in the reported cohort had higher burden of chronic illness, including conditions (e.g., diabetes, renal failure) associated with chronic inflammation and higher risk of sepsis. What are the hospital mortality outcomes in Germany in rural vs urban populations for other acute conditions?
Minor
1. Line 51: the refence number should be changed to 12.
2. Lines 108 and 119: use consistent terminology throughout the manuscript: either “focus of infection” or “sepsis focus”.
3. Table 3: limit estimates to a single digit post decimal point to improve readability. The extra digital data does not add to inference.
4. Figure 2: the displayed data duplicate that for Table 3. An option would be to remove ORs form the table. Add 95% CIs to the figure data.
Author Response
Point to point reply
The manuscript describes the short- and long-term case fatality of rural vs urban septic patients in a large population study from Germany. The topic is timely and of substantial importance for clinical care and health policy efforts to reduce sepsis-related outcome disparities. The manuscript is well-written. However, there are several areas of concern.
Major comments
- The reported cohort includes septic patients managed nearly 10 years ago. Progress in patient care and potential population/environmental changes may limit the generalizability of reported findings to contemporary rural vs urban sepsis outcomes. This consideration should be included under study limitations.
Reply: We thank the reviewer for the comments and added the actuality of data as limitation of the study in the discussion, which reads as follows in the revised version:
Page 12, lines 292-295: “Fifth, the sepsis cohort was identified in the years 2013 and 2014. Possible medical and environmental changes may limit the applicability of the findings to current care settings and underscore the need to replicate this study in the context of current research.”
- Lines 54-55: the cited study by Goodwin et al had indeed similar case fatality among septic patients living in medically underserved areas vs those not residing in such areas; however, both crude and adjusted mortality rates (which reflect better a disease-specific population burden) were markedly higher among the former.
Reply: Given the focus of our study on urban-rural disparities in sepsis case fatality, we highlighted the finding of Goodwin et al. that case fatalities did not differ between patients residing in medically underserved areas vs. those not residing in such areas in our manuscript. In the light of these results, differences in mortality observed by Goodwin et al. may be mainly driven by differences in sepsis incidence between areas and associated risk factors, which lay outside the focus of our study which bases on a cohort of hospital-treated sepsis patients and assesses the association between their place of living and sepsis case fatality.
- Lines 108-110, 118-120, and 289-290: the authors collected data on type of organ dysfunction as well as on organ support (mechanical ventilation and hemodialysis). However, the latter 2 categories were not included in the adjusted models, as noted in lines 118-120, with this selection confirmed in the Discussion (lines 289-220). Illness severity is a key confounder in examination of patient outcomes and would be expected to affect the estimates of the association of rural vs urban residence with morality in sepsis. Because physiological data, commonly used in severity of illness scores based on clinical records, may not be available in administrative data, investigators often use organ dysfunction data and organ support interventions as proxies for severity of illness sin adjusted models. For example, a recent report by Bosch NA, et al (AnnalsATS 2022; 19(6): 1072-1076) demonstrated similar accuracy in predicting hospital mortality between the number of organ dysfunctions and SOFA scores. Model findings for in-hospital- and 12-month mortality overall and in the examined subgroups, adjusted for measure of illness severity should be reported. Summary data on the number of organ dysfunctions should be added to Table 1 for better perspective on the cohort.
Reply: Thank you very much for this important comment. Indeed, we did not include characteristics of the sepsis, including the type of organ dysfunction and organ support (mechanical ventilation and hemodialysis) in the adjustment model for good reasons. In our study, we compared rural and urban sepsis patients. Hence, the grouping variable of interest (i.e., urban versus rural) indicates, whether the patients live in urban or rural area before the onset of sepsis. All statistical theories on causality including Rubin’s causal framework and graphical theories of Pearl (i.e., directed acyclic graphs) have shown that adjusting for “posttreatment” variables leads to bias estimates (i.e, posttreatment bias due to conditioning on posttreatment variables, see [1, 2]). In our case there is no classical treatment variable with a treatment and a control group. Instead, we focus on a grouping variable and adjusted comparisons of the two groups of rural and urban patients. Considering the temporal order, only those variables that are temporally prior or equal the grouping variable can be possible confounding variables. Characteristics of the sepsis including the severity of sepsis do not fulfill this temporal criterion, as that severity can be a consequence of living in a rural area. Imagine that availability of immediate medical assistance has important impact on the course of the sepsis and may lead to more or less severe sepsis. Do also imagine that rural and urban areas differ in the availability of immediate medical assistance, which implied differences in the severity of sepsis between rural and urban patients. Hence, sepsis severity would be a consequence of living in a rural versus urban area in this scenario. If one were to adjust for sepsis severity (i.e., a post-“treatment” variable) a (downward) bias would result for all group comparisons, regarding all outcomes that depend on sepsis severity (including mortality). This would be a classical example of a post-“treatment” bias. In order to avoid such biases we did not include any variables that characterize the kind and number of organ dysfunctions or treatment during the acute disease.
Nevertheless, we compared the numbers of organ dysfunctions between rural and urban patients, which hardly differed between the two groups (Urban; M = 1.11, SD = 1.26; Rural M = 1.06, SD = 1.186). We added the number of organ dysfunctions in Table 1.
- How many patients in each group were admitted to ICU overall and among those with severe sepsis?
Reply: Among urban and rural sepsis patients, 27.7% and 24.8% were admitted to ICU, respectively (severe sepsis urban: 47.3% [46.7%, 47.8%]; severe sepsis rural: 44.1% [43.2%, 44.9%]). This information was also added to Table 1.
- Lines 115-121: The risk-adjustment covariates used in modeling should be moved to the statistical analysis subsection by the relevant models for clarity.
Reply: We followed the suggestion of the reviewer and moved the list of covariates to the paragraph: statistical analyses.
- Shorten the Results section and limit duplication of Tabular data in the narrative.
Reply: We shortened the Results section to present our results clearer and reduced duplicate information.
- Lines 238-246: The implied comparable access to care between rural and urban populations in Germany does not explain the prognostic advantage of septic rural patients and would rather suggest an otherwise similar outcome.
Reply: We agree with the reviewer that access to care does not serve for a reasonable explanation and modified the structure of the discussion accordingly. In the revised version we aim at understanding our results in the context of published empirical findings regarding other critical conditions where urban/rural disparities were found, such as stroke or myocardial infarction. Hence, there is empirical evidence that supports the hypothesis that access to care plays a role for the outcome of acute medical conditions, but may be differential comparing sepsis to other acute medical illnesses. We argue that early sepsis treatment rests at the timely start of an adequate antibiotic and fluid therapy and does mostly not require specified treatment units such as stroke units or cardiac catheter laboratories. We added this point in the discussion of the paper. See page 12, second paragraph:
“Above these environmental factors, early sepsis treatment rests at the timely start of an adequate antibiotic and fluid therapy and does mostly not require specified treatment units such as stroke units or cardiac catheter laboratories. Therefore, rural residence may not put patients at disadvantage regarding their emergency care and may explain differences to urban-rural-disparities found in stroke or myocardial infarction patients in Germany. Furthermore, […]”
- Lines 253-254: the referenced study #25 is cited by the authors to indicate that in Germany there are “inverse associations between the number of hospital beds and sepsis outcomes” to support their postulate that septic patients treated in tertiary centers and university hospitals (presumably mostly urban patients) would have higher mortality and thus would explain in part the lower case fatality among rural patients. However, the quoted statement indicates the opposite, that is, that higher number of beds would be associated with lower mortality. More critical, however, is that the cited study by Engel et al showed no association between the number of beds or university hospitals and mortality on adjusted analyses. Thus, the prognostic advantage of rural septic patients in the present study Is not explained by the Discussion in lines 247-264. In addition, as noted by the authors in lines 262-264, some septic rural patients may have actually worse outcome when seeking treatment in tertiary centers, which would lead to the opposite effect than that observed in the present study, at least for some rural patients. The authors acknowledge the lack of data on hospital characteristics in their data under study limitations.
Reply: We apologize, citation 25 was misplaced. Instead of the publication by Engel et al., we discussed the results of an analysis of nationwide DRG data which was linked to data on hospital characteristics of the mandatory structured quality report of hospitals in Germany published as abstract in Crit Care [3]. However, given the lack of data on that topic in Germany apart from the beforementioned abstract, we decided to omit this section from the discussion in the revised discussion.
- Lines 265-286: the authors postulate higher burdens of environmental risk factors among urban residents as an explanation for the lower mortality among septic rural residents, including among others, higher levels of chronic inflammation. If these are key factors, rural residents would be expected to have better outcomes for other acute conditions. This is, however, not the case, as the authors cite worse outcomes of rural patients for other acute conditions (lines 44-45) in other countries (mostly high-income ones), which appear at odds with the proposed argument. Notably, the rural residents in the reported cohort had higher burden of chronic illness, including conditions (e.g., diabetes, renal failure) associated with chronic inflammation and higher risk of sepsis. What are the hospital mortality outcomes in Germany in rural vs urban populations for other acute conditions?
Reply: We added estimates for urban-rural case fatality disparities in Germany and discussed our results in this context. We hypothesized that the rural survival deficit, which has been demonstrated for sepsis but not for myocardial infarction or stroke, is partly influenced by environmental factors and partly by the fact that specialized treatment facilities such as stroke units or cardiac catheterization laboratories do not necessarily need to be available for the initial treatment of ambulatory-acquired sepsis. We hope that the revised discussion better emphazises these hypotheses. To deepen our understanding of the underlying mechanisms, further research from both epidemiological and molecular research is needed.
Minor
- Line 51: the refence number should be changed to 12.
Reply: We changed the reference and apologize for this mistake.
- Lines 108 and 119: use consistent terminology throughout the manuscript: either “focus of infection” or “sepsis focus”.
Reply: We harmonized the terminology to focus of infection.
- Table 3: limit estimates to a single digit post decimal point to improve readability. The extra digital data does not add to inference.
Reply: We decided to limit estimates to two digits after the decimal point, as using one digit post decimal may lead to limited informative confidence intervals in some cases.
- Figure 2: the displayed data duplicate that for Table 3. An option would be to remove ORs form the table. Add 95% CIs to the figure data.
Reply: We agree that the information presented in Figure 2 refers to the OR in Table 3. However, we feel that the graphical representation gives a better visual impression of the result pattern, whereas the numbers in Table 3 provide information that is more detailed. Depending on the preferences and the further use of the presented information by the readership (e.g., for latter metaanalysis) tables and figures can be valuable. Therefore, we decided to keep Table 3 as well as Figure2.
References:
- Rosenbaum, P.R., The Consquences of Adjustment for a Concomitant Variable That Has Been Affected by the Treatment. Journal of the Royal Statistical Society. Series A (General), 1984. 147(5): p. 656-666.
- Montgomery, J.M., B. Nyhan, and M. Torres, How Conditioning on Posttreatment Variables Can Ruin Your Experiment and What to Do about It. American Journal of Political Science, 2018. 62(3): p. 760-775.
- Schwarzkopf, D., et al., Effects of structural hospital characteristics on risk-adjusted hospital mortality in patients with severe sepsis – analysis of German national administrative data. Critical care (London, England), 2019. 23.
Reviewer 2 Report
Matthaeus-Kraemer et al., conducted a cohort study of sepsis in German with the aim of comparison between urban and rural incidences. They found that rural residence is associated with a short-term and long-term survival benefit in patients with community-acquired sepsis though they had hypnotized that rural habitants might have less facilities of care and thus are prone to fatality risks.
Some issues are unclear to me, first, the sepsis identification is not well presented and the method section must be refined in this regard. Furthermore, apart from rural and urban residency, did the authors evaluate any other factors contributing to the risk of sepsis and its mortality? Some data are provided in Tab.1 but the results are not discussed well.
The data mainly have focused on statistical analysis which make the riding difficult and impaired. I suggest that the statistical results be less discussed ant the mail findings be more detailed highlighting the importance of the findings and the benefits of them.
Some grammatical errors are through the text such as:
Line 11: In line with these observations, an increased population-level sepsis incidence and mortality was observed in medically…, plural subject and singular verb!?
Line 54: no differences…. Must change to “No difference or there were not any differences”.
Author Response
Reviewer 2
Matthaeus-Kraemer et al., conducted a cohort study of sepsis in German with the aim of comparison between urban and rural incidences. They found that rural residence is associated with a short-term and long-term survival benefit in patients with community-acquired sepsis though they had hypnotized that rural habitants might have less facilities of care and thus are prone to fatality risks.
Some issues are unclear to me, first, the sepsis identification is not well presented and the method section must be refined in this regard.
Reply: We apologize for the lack of clarity and added the following explanation of the methods section: “In 2013/2014, sepsis was defined according to the sepsis-1/2 criteria as systemic inflam-matory response syndrome due to infection [16] in Germany and comprised patients with and without septic organ dysfunction (patients with and without severe sepsis according to the sepsis-1/2 definition). Explicit sepsis codes comprised clinical sepsis codes (“R codes”) and microbiological sepsis codes (e.g. “A codes”, “B codes”). Severe sepsis was defined by codes R56.1 (severe sepsis) and R57.2 (septic shock).”
Furthermore, apart from rural and urban residency, did the authors evaluate any other factors contributing to the risk of sepsis and its mortality? Some data are provided in Tab.1 but the results are not discussed well.
Reply: We thank the reviewer for this question. The focus of this manuscript was on urban-rural disparities for sepsis. For that reason, we did not evaluate other factors contributing to the occurrence or case fatality of sepsis. We did focus on other risk factors in previous analyses, e.g. Spoden et al., Frontiers 2022 (https://doi.org/10.3389/fmed.2023.1137027)
The data mainly have focused on statistical analysis which make the riding difficult and impaired. I suggest that the statistical results be less discussed ant the mail findings be more detailed highlighting the importance of the findings and the benefits of them.
Reply: We shortened the results section in order to avoid duplicate information with Tables and Figures. We also highlighted the importance of findings in the conclusion, which now reads as follows:
Conclusion: " Rural residence is associated with a short-term and long-term survival benefit in patients with community-acquired sepsis. Given that our world is getting increasingly urbanized with an estimated 7 of 10 people in the world that will live in urban areas by 2050 [1], and the United Nations (UN) identified urbanization as one of four “demographic mega-trends” [2], rurther research on patient, community, and health care system factors that distinguish rural from urban patient populations is needed to understand the underlying mechanisms. This may also help to identify preventive measures to reduce the case fatality of sepsis and its long term impact on patients, relatives and the society”
We decided to refrain from shortening the methods section, was from our point of view, it is important to give a detailed description of the methodology to support the interpretation of results and to allow the replication of the study method.
Some grammatical errors are through the text such as:
Line 11: In line with these observations, an increased population-level sepsis incidence and mortality was observed in medically…, plural subject and singular verb!?
Reply: We changed this grammatical error.
Line 54: no differences…. Must change to “No difference or there were not any differences”.
Reply: We corrected this sentence.
References:
- Group, W.B., Demographic Trends and Urbanization. 2021, World Bank.
- Zhongming, Z., et al., Shaping the Trends of Our Time, in Report of the UN Economist Network for the UN 75th Anniversary. 2020, United Nations.
Round 2
Reviewer 1 Report
The authors have provided their rationale for not including measures of illness severity in their modeling (comment 3). Adding an abbreviated comment on their rationale to the statistical analysis will enhance readers' inference of the study findings.
Author Response
Dear Reviewer,
we thank you for this suggestion. We added the following explanation to the discussion:
"It is important to note that the temporal order needs to be considered in the selection of covariates for adjusted group comparisons. Only covariates that are temporally prior or equal to the grouping variable of urban or rural residence can be confounding variables. Characteristics of the sepsis including the severity of sepsis, organ dysfunction and measures of organ support do not fulfill this temporal criterion. Persons’ living environment exists before the onset of the sepsis. The severity and the course of the sepsis can be a consequence of living in an urban or rural area but cannot be a confounder. Adjusting for posttreatment variables leads to biased parameter estimates [18]. Therefore, organ dysfunction and measures of organ support, for example, were not included in the adjustment model."
Kind regards,
Claudia Matthäus-Krämer